

# Observing atmospheric rivers using GNSS radio occultation data

Bahareh Rahimi[1], Ulrich Foelsche[1,2]

[1]Institute of Physics, Department of Astrophysics and Geophysics (AGP), University of Graz, Austria
[2] Wegener Center for Climate and Global Change (WEGC), University of Graz, Austria

*Correspondence to*: Bahareh Rahimi (bahareh.rahimi@uni-graz.at)

**Abstract.**

Atmospheric Rivers (AR) are comparatively narrow regions in the atmosphere that are responsible for most of the horizontal
transport of water vapor in the extra tropics, which are responsible for many extreme precipitation events and floodings at mid-
latitudes, including Europe and the US. The critical role of ARs in global moisture transport and precipitation dynamics
necessitates accurate water vapor measurements for both understanding and forecasting these phenomena. While the integrated
water vapor content (IWV) of ARs can be well measured with microwave and infrared sounders, the vertical structure is less
well known. In this study, we analysed if specific humidity profiles and IWV values from Global Navigation Satellite System
Radio Occultation (GNSS-RO) measurements provide additional information for the study of ARs, in particular regarding
their vertical structure. The retrieval of water vapor from GNSS-RO data requires background information, which is usually
incorporated by the one-dimensional variational method (1D-Var) that combine observations and background in an optimal
manner. We compared data from the COSMIC Data Analysis and Archive Centre (CDAAC), operated by the University
Corporation for Atmospheric Research (UCAR) in Boulder, Colorado with data from the Wegener Center for Climate and
Global Change (WEGC) at the University of Graz, Austria. We found that retrievals from both centres agree very well in the
altitude range, where the 1D-Var weights the observations strongly, even if the employed background profiles are very
different. This demonstrates that GNSS-RO data provide indeed additional vertically-resolved information, which was not
already contained in the background or in operational analyses. IWV values from CDAAC and WEGC agree generally very
well, however, both tend to underestimate the values obtained by Special Sensor Microwave Imager/Sounder (SSMI/S) data,
since GNSS-RO profiles not always reach the lowermost part of the atmosphere, leading to a systematic bias in the IWV data,
which decreases with better penetration characteristics of the GNSS-RO data. The results suggest that is promising to combine
the GNSS-RO data – with very high vertical resolution with SSMI/S data – with high horizontal resolution to get a more
compete view of the 3D structure of ARs.





## 1 Introduction

The monitoring of global atmospheric water vapor is crucial for accurate weather prediction and understanding the dynamics of water vapor transport (WVT). Latent heat, released during condensation and absorbed when liquid water evaporates, is a primary driver of atmospheric processes. This underscores the essential role of water vapor in thermal energy transmission and the atmospheric hydrologic cycle (e.g., Businger et al., 1996; Emardson et al., 1998). Integrated water vapor (IWV), often expressed as precipitable water (PW), is crucial for environmental models.

Zhu and Newell (1998) proposed that the majority of water vapor transport across midlatitudes takes place through elongated features in the lower troposphere, known as atmospheric rivers (ARs). Generally, ARs are characterized by widths of approximately 1000 km, sometimes appearing narrower in specific scenarios. Their lengths usually extend to about 2000 km but can be longer under certain atmospheric conditions. Ralph et al. (2004) further refined the understanding of ARs, defining them as narrow corridors of water vapor that exceed 2000 km in length, are no wider than 1000 km, and have a PW of $\geq 20$ mm

(corresponding to IWV $\geq 20$ kg/m$^2$). This precise definition was later employed by Neiman et al. (2008b). Despite covering only about 10% of the Earth's circumference at specific latitudes, ARs are responsible for more than 90% of the meridional transport of sensible and latent heat from the (sub)tropics to mid-latitudes (Zhu and Newell, 1998; Ralph et al., 2004)

ARs have been associated with extreme weather events, including flooding and heavy precipitation across various regions, causing significant damage (e.g., Ralph et al., 2003, 2006; Stohl et al., 2008). Given the challenge of analysing and predicting

ARs due to the scarcity of traditional meteorological observations over oceans, the optimal utilization of satellite data has been shown to enhance the accuracy of numerical weather prediction models (e.g., Ralph et al., 2004, 2006; Neiman et al., 2008b). Since the 1990s, the number of observation techniques for measuring IWV has increased, with modern techniques providing high temporal resolution. Global Navigation Satellite System (GNSS) radio occultation (RO), radiosonde, and microwave radiometer measurements of IWV have shown reasonable agreement (Bouma and Stoew, 2001; Güldner, 2001; Dai et al.,

2002). PW derived from the Special Sensor Microwave Imager (SSM/I) is key for monitoring ARs over oceans (Ralph et al., 2004; Neiman et al., 2008b). The Special Sensor Microwave Imager/Sounder (SSMI/S) employed by the Defence Meteorological Satellite Program (DMSP) provides global coverage over the oceans for PW, with uniform sampling and mean errors under 0.5 mm (Xue et al., 2019). However, SSMI/S data are influenced by heavy rain and complicated by the large and highly variable emission from land, limiting its application to oceanic areas (Elsaesser and Kummerow, 2008; Schluessel and

Emery, 1990; Wentz and Spencer, 1998).

Significant progress has been made in the field of numerical weather prediction (NWP) over recent decades (Bauer et al., 2016; Alley et al., 2019). However, accurately predicting AR intensities and trajectories continues to pose significant challenges. For instance, the ensemble prediction system of the European Centre for Medium-Range Weather Forecasts (ECMWF) shows that, on average, only 75% of its ensemble members can forecast the AR landfall location within a 250 km radius with a two-day

lead time, which decreases to below 25% for forecasts extending to five days (DeFlorio et al., 2018). A crucial aspect of enhancing AR forecasts involves refining atmospheric state analyses. These analyses are pivotal as they provide the initial



conditions for NWP models. Given the atmospheric system's inherent deterministic chaos, any initial inaccuracies can amplify over time (Lorenz, 1969). This is especially true for global ocean regions, where direct observations are limited (e.g., Ota et al., 2018). The role of accurately observing atmospheric moisture becomes even more critical over these areas, as studies using

adjoint models have demonstrated that the precision of short-term precipitation forecasts for ARs making landfall is highly dependent on the initial moisture estimates within and surrounding the ARs (Doyle et al., 2014; Stone et al., 2020; Demirdjian et al., 2020; Reynolds et al., 2019).

The GNSS-RO limb-sounding technique relies on GNSS radio signals that are refracted and delayed by the atmospheric refractivity field during their propagation to a receiver on a Low Earth Orbit (LEO) satellite. With the satellites' relative

movements, the atmosphere is scanned vertically, providing excellent vertical resolution. Observations obtained from this technique are available in nearly all-weather conditions, as signals in the L-band microwave range are unaffected by clouds, facilitating a seamless observation record without the need for intercalibration or temporal overlap between different missions (Foelsche et al., 2011a; Angerer et al., 2017).

The vertical resolution of GNSS RO ranges from approximately 100 m in the lower troposphere to about 1 km in the

stratosphere (Kursinski et al., 1997; Gorbunov et al., 2004), with variations identified by Zeng et al. (2019) based on the specific atmospheric layers and latitudes under observation. GNSS-RO thus captures high-vertical-resolution profiles of atmospheric bending angle and refractivity, which correlate directly to air density under dry atmospheric conditions.

For moist conditions in the troposphere, the retrieval requires a priori information. GNSS-RO data have therefore been primarily used for accurate monitoring of atmospheric temperature in the upper troposphere and lower stratosphere (e.g.,

Steiner et al., 2001; Foelsche et al., 2008a)

However, the potential for observing water vapor in the (lower) troposphere is increasingly recognized (e.g., Kursinski et al., 1995; Bouma and Stoew, 2001; Rieckh et al., 2018) and GNSS-RO humidity data have even proven to be valuable under particularly dry conditions (Rieckh et al., 2018) and in the stratosphere – in the special situation after the Hunga Tonga eruption (Randel et al., 2023). GNSS-RO data have already been successfully used to observe ARs (e.g., Neiman at el., 2008a: Murphy

and Haase, 2022), and the assimilation of GNSS-RO data has been shown to improve AR forecasts (e.g., Ma et al., 2011).

Both GNSS-RO and SSMI/S observations offer valuable insights, particularly over the ocean, and complement each other in various ways. The SSMI/S data can observe the IWV content, offering consistent horizontal information regarding the overall moisture content. However, they do not provide any details about the vertical distribution of moisture within the atmosphere. On the other hand, GNSS-RO measurements offer the ability to retrieve vertical profiles of atmospheric variables, such as

moisture, with a high vertical resolution of approximately 200 m in the troposphere.

Despite their effectiveness in vertical profiling, GNSS-RO data have a limitation in their horizontal resolution. This limitation means that GNSS-RO data may not accurately capture small-scale horizontal variations in atmospheric moisture. Therefore, in applications where understanding both vertical and horizontal moisture distributions is essential, it might be beneficial to use GNSS-RO and SSMI/S data in tandem.



In this study, we analyse if GNSS-RO data provide additional water vapor information within and in the vicinity of ARs, given the fact that a-priori information is needed to derive humidity profiles. Therefore, we focused on investigating the impact of background water vapor information on the GNSS RO-derived humidity profiles. We aim to discern whether the variations in background water vapor datasets can introduce significant discrepancies in the final moisture profiles and, if so, to what extent, thereby shaping our understanding of atmospheric moisture profiles obtained through RO techniques.

This study is organized as follows: Section 2 and 3 provide an overview of the data and methodology used in this study. Section 4 discusses the results in detail, and Section 5 summarizes the key findings and suggests future research directions.

## 2 Data

### 2.1 GNSS-RO and 1D-Var methodology for atmospheric profiling

Since its operational inception in 2001, the GNSS-RO technique has significantly enhanced atmospheric profiling by providing
high-resolution data on temperature, water vapor, and pressure. This advancement was built upon the foundational work of the 1995 GPS/MET mission, which pioneered the use of GNSS for atmospheric sounding, albeit with limitations in capturing detailed wet parameters at lower altitudes (Ware et al., 1996; Kursinski et al., 1997; Rocken et al., 1997; Steiner et al., 1999; 2001). Our study employs refined GNSS-RO data that overcome most of these historical limitations.

The GNSS-RO technique, which measures the phase delay of radio waves emitted by GNSS satellites, has become pivotal for
capturing accurate vertical atmospheric profiles (e.g., Rieckh et al., 2018). Despite facing challenges in data accuracy at lowermost altitudes and a resolution of approximately 200 km in the lower troposphere (Healy et al., 2014; Scherllin-Pirscher et al., 2011), technological advancements in signal processing and data inversion have bolstered its reliability and precision, making the GNSS-RO technique a formidable tool in atmospheric research and analysis.

Occultation events, lasting 1-2 minutes, result in the bending and delay of signals due to atmospheric density gradients. The
vertical refractivity profiles are derived using accumulated bending angles and observed phase data involving GNSS and LEO satellite orbits (Melbourne, 1994). The Abel transform allows to retrieve atmospheric refractivity profiles from observed bending angles (Kursinski et al., 1997; Hajj et al., 2002). Once refractivity is obtained, atmospheric parameters like density, pressure, temperature, and water vapor pressure are derived using the Smith-Weintraub equation, the hydrostatic equilibrium principle and the equation of state (Kursinski et al., 1995; Smith and Weintraub 1953; Aparicio and Laroche, 2011).

In dry conditions, refractivity depends only on air density. "Dry temperature", derived from GNSS-RO profiles by neglecting the water vapor contribution to refractivity, differs from physical temperature by less than 0.1 K above ~15 km in the Topics and above 8 km in high latitude winters (Foelsche et al., 2008a, Danzer et al., 2014).

At lower altitudes, the presence of significant water vapor necessitates the use of additional background data to derive either physical temperature or specific humidity – or both (Kursinski et al., 1997). Two main methods exist for humidity derivation:
direct retrieval and One-Dimensional Variational (1D-Var) retrieval. The direct method, once popular for moist air retrieval algorithms (Kursinski et al., 1995; Ware et al., 1996), uses climatological temperature information to retrieve atmospheric





profiles but can introduce uncertainties from assumed background data. In the 1D-Var method, a cost function, $J(x)$, is minimized, where:

$$J(\mathbf{x}) = \frac{1}{2}(\mathbf{x} - \mathbf{x}_b)^T \mathbf{B}^{-1}(\mathbf{x} - \mathbf{x}_b) + \frac{1}{2}(\boldsymbol{y}_0 - H[\mathbf{x}])^T \mathbf{O}^{-1}(\boldsymbol{y}_0 - H[\mathbf{x}]) \tag{1}$$

where $\mathbf{x}$ is the state vector representing the atmospheric parameter being estimated (e.g., temperature or humidity). $\mathbf{x}_b$ is the background state. $\mathbf{B}$ and $\mathbf{O}$ are error covariance matrices related to the background state and observation, respectively. $\boldsymbol{y}_0$ is the observation. $H[\mathbf{x}]$ is a forward operator mapping from state $\mathbf{x}$ to the observation space.

The use of this method results in an optimally estimated atmospheric state profile, providing a solid foundation for various meteorological applications by taking into account the uncertainties of the observations and of the background.

Different implementations of the 1D-Var method utilize RO observations along with prescribed background data. The Constellation Observing System for Meteorology, Ionosphere, and Climate (COSMIC) Data Analysis and Archive Centre (CDAAC), operated by the University Corporation for Atmospheric Research (UCAR), employs the one-dimensional variational data assimilation (1D-Var) method for RO moist profiles. This 1D-Var method is utilized as there is significant humidity content. The presence of substantial moisture makes it impractical to directly use temperature measurements for retrieving wet parameters. Hence, the 1D-Var approach facilitates the extraction of these parameters in regions with high humidity. The Occultation Processing System (OPS) at the Wegener Centre for Climate and Global Change (WEGC) has also implemented a simplified linearized 1D-Var method since 2013, with reliable results reported in several studies (e.g., Li et al., 2020). The accuracy and reliability of methods like those employed by CDAAC and WEGC hinge on the precise estimation of error correlations and the uncertainties in both the observational data and the background data, thereby shaping the quality of the retrieved moist atmospheric profiles (Li et al., 2020).

### 2.1.1 WEGC profiles

The WEGC provides RO level-2 profiles, which include both moist and dry atmospheric profiles from an altitude of 0.1 km to 80 km with a vertical resolution of 0.1 km. For this study, we utilized level-2 moist profiles from WEGC to derive specific humidity profiles and calculate IWV values, specifically employing the latest version of WEGC data from the Occultation Processing System version 5.6 (OPSv5.6). WEGC utilizes spatially interpolated ECMWF forecast fields, 24 or 30 hours in advance, as background data at the RO profile locations, referred to henceforth as ECMWF-b (Schwärz et al., 2016). Post-retrieval, a quality flag, using the ECMWF analysis data as a reference (ECMWF-r), is generated to evaluate the temperature and humidity profiles. These reference data do not alter the retrieved profiles but serves as a reference for quality assessment. Additionally, WEGC has developed a linearized version of the 1D-Var method to generate moist profiles (Kirchengast et al., 2010), relying mainly on the RO-retrieved dry temperature, with uncertainties modelled using an empirical error model (Scherllin-Pirscher et al., 2011).

GNSS-RO data provide altitudinal resolved information, but the profiles are not vertical and the tangent point (TP, the point of closest approach of the ray) travels significant horizontal distances during an occultation event (Foelsche et al., 2011b).





Within the OPS retrieval framework, the TP is calculated assuming straight line propagation of the GNSS signals, and the mean TP (the "location of the profile") is defined as the point where the straight line between LEO and GNSS satellite would

hit the Earth's surface, corresponding to an altitude of 10 km to 15 km, depending on atmospheric density variations. This mean TP can be computed by just knowing the orbit parameters of the GNSS and LEO satellites, which allows to predict GNSS-RO event locations. It is well suited to describe the profile location for temperature information in the upper troposphere and lower stratosphere, but it does not represent the profile in the lower troposphere very well. This mean TP location is pivotal, since it is used to extract co-located profiles from other datasets, which provide essential background information for

the retrieval process and for validating the profiles post-retrieval.

### 2.1.2 CDAAC Profiles

Level-2 Profiles from CDAAC can be found and downloaded at the CDAAC website (https://cdaac-www.cosmic.ucar.edu/). The 'wetPrf' file type encompasses the retrieved temperature, pressure, and water vapor partial pressure profiles, sampled every 0.1 km of altitude from 0.1 km up to 40 km. Notably, data from GNSS-RO above 40 km are generally not applicable to

compute wet profiles due to the upper atmosphere's extremely dry conditions, where water vapor is typically negligible or absent. For this study, the latest reprocessed version, 2013.2350, was utilized for its enhanced accuracy and data integrity.

The wet profiles retrieved are collocated with various analytical datasets, including ERA-40 Interim reanalysis data (ERA), European Centre for Medium-Range Weather Forecasts profiles (ECMWF), National Centres for Environmental Prediction operational analysis (NCEP), and the Global Forecast System data (GFS). These collocated profiles are accessible at CDAAC

within the 'eraPrf', 'echPrf', and 'gfsPrf' products. Predominantly, ERA data serve as background for CDAAC's 1D-Var retrieval process. The 'eraPrf' type is matched with the RO profile and serves as the initial guess for moisture determination below 10 km and for comparison in post-processing. Should ECMWF profiles be utilized as the background for RO retrieval, it will be explicitly stated; however, the ERA-Interim dataset is typically the preferred source.

At CDAAC, the occultation point, i.e., the point on the Earth's surface to which the retrieved profiles are assigned, is estimated

as the TP of the ray connecting GNSS and LEO for which the excess phase of the L1 signal is equal 500 m. This, on average, corresponds to 3-4 km altitude (https://cdaac-www.cosmic.ucar.edu/cdaac/doc/documents/roam05.doc). This definition of the occultation point differs considerably from the one employed by WEGC (see section 2.1.2) and it is better suited to represent the profile in the lower troposphere. As a consequence, the background (and reference) profiles at CDAAC and WEGC are extracted at quite different horizontal locations. This complicates a direct comparison of the results, however, it allows for a

determination of the impact of the background – in particular in case of strong horizontal variability, where the background profiles then represent very different atmospheric conditions (see section 4).





## 2.2 SSMI/S data

SSMI/S observations are indispensable for continuous monitoring of atmospheric and oceanic phenomena such as integrated water vapor, temperature, and wind speed, offering comprehensive global coverage (NESDIS STAR, 2024). The instruments
have a swath width of about 1400 km, allowing for extensive surveillance of Earth's surface. They simultaneously measure thermally emitted radiation across 24 channels, from 19 to 183 GHz, in a conical scanning mode. This multifaceted approach enables detailed observations of brightness temperatures at both microwave temperature and water vapor sounding channels, as well as imager channels, from a single scan angle, thus enhancing the analysis of atmospheric parameters including IWV. The observations are primarily performed over oceans where the retrieval algorithms for parameters like IWV are optimized
for the marine environment. This focus is due in part to the more uniform and predictable emissivity of ocean surfaces compared to land, where varying emissivity can introduce retrieval challenges and reduce measurement accuracy. Hence, while SSMI/S data encompasses the global surface, analyses and forecasts primarily leverage oceanic data for its reliability. For superior data quality, users can turn to the gridded DMSP SSM/I and SSMIS ocean data products from the Remote Sensing Systems (RSS). These products compile oceanic information across various timescales, providing an invaluable tool for those
needing to analyse and visualize water vapor content. Enhanced data access is facilitated through Python plotting routines, as detailed in RSS data recipes, which assist in managing, plotting, and interpreting the stored values within these gridded, time-averaged oceanic datasets. This research incorporates data from the SSMI/S F16 and F17 satellites, launched on October 18, 2003, and November 4, 2006, respectively. The all-sky daily RSS ocean product for these satellites is accessible for download at the RSS website (https://www.remss.com/missions/ssmi/).The horizontal resolution of SSMI/S data for IWV retrievals is
generally between 25 and 50 km, while the vertical resolution is limited due to the nature of passive microwave radiometry, which yields integrated measurements of atmospheric parameters. Consequently, the SSMI/S IWV dataset represents the total water vapor content in the atmospheric column.

## 3 Methodology

### 3.1 Selected AR events

In this study, we screened dozens of AR events covered by GNSS-EO data, and we selected six different AR events to analyse and compare the retrieved moist profiles in detail. The selection of these events was based on several criteria. First, we captured the diversity of latitudinal regions in both hemispheres to reflect the influence of background water vapor on the retrieved profiles. Second, the AR events should fall into the period after 2007, when the COSMIC-1 constellation, consisting of 6 LEO satellites, started to provide dense observations (2007 to 2010) and cover other satellites such as Meteorological Operational
Satellite Programme (MetOp) A, B, C, Gravity Recovery and Climate Experiment (GRACE), TerraSAR-X (TSX), and the six satellites of the COSMIC-2 constellation. For each event we defined a rectangular latitude-longitude domain, in order to also cover areas outside of the AR with a wide range of IWV values. The study domains of these six AR events are shown in Figure





1, and detailed information on the study domains, period of events, and the available LEO satellites is shown in Table 1. The designations of the events refer to the area, where the effects of the AR where most pronounced.


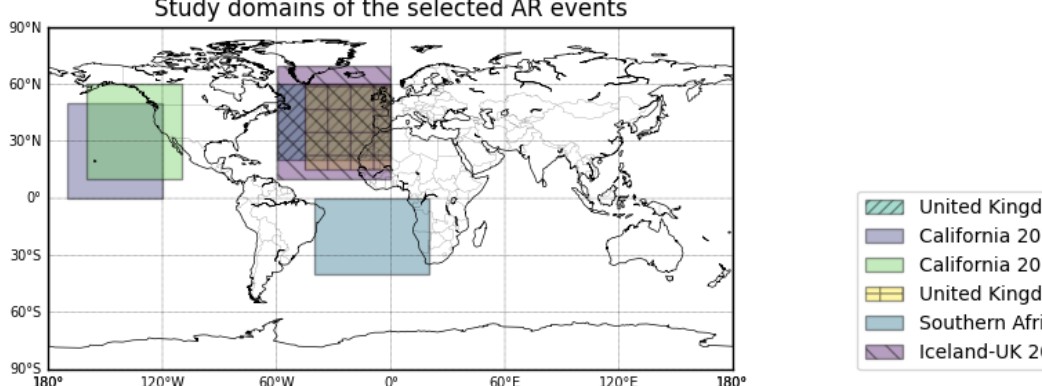

**Figure 1**. Study domain locations of the six selected AR events

**Table 1**. Detailed information on six selected AR events for comparison.

| Region | Date of the event | Study domain location (Latitude °/ Longitude °) | Present satellites |
|---|---|---|---|
| United Kingdom | 23-25 Feb 2020 | 20 to 60 / 0 to -60 | COSMIC-1, COSMIC-2, MetOp-A, MetOp-B, and MetOp-C |
| California | 13-14 Feb 2019 | 0 to 50 / - 120 to -170 | COSMIC-1, MetOp-A, and MetOp-B |
| California | 7-9 Feb 2017 | 10 to 60 / - 110 to -160 | COSMIC-1, MetOp-A, MetOp-B, and GRACE |
| United Kingdom | 3-5 Dec 2015 | 10 to 70 / 0 to -60 | COSMIC-1, MetOp-A, MetOp-B, GRACE, and TSX |
| Southern Africa | 26-27 Sep 2009 | 0 to -40 / 20 to -40 | COSMIC-1, MetOp-A, GRACE, and TSX |
| Iceland- UK | 10-14 Oct 2009 | 15 to 75 / 0 to -45 | COSMIC-1, MetOp-A, GRACE, and TSX |


As an example, Fig. 2 illustrates the Iceland-UK AR event in 2009. During this event, four GNSS-RO satellites, namely COSMIC-1, MetOp- A, TSX, and GRACE, were operational and provided data for our analysis. The AR event spanned five days, starting on October 10, 2009, and ending on October 14, 2009. To capture the temporal evolution of the AR event, we created four separate plots, each corresponding to a specific day within the event period, excluding the first day (to save space).

The figure showcases the spatial distribution of the IWV values (in kg m$^{-2}$) obtained from the SSMI/S measurements, providing insights into the moisture content along the AR path. In addition, we have included all available GNSS-RO profiles within the study domain, where the colours of the circles represent different LEO satellites/constellations. While the figure showcases the horizontal moisture distribution derived from the SSMI/S observations, it juxtaposes this with the RO data points. From





these RO data points, IWV can also be calculated for their corresponding locations. In Figure 2, the white regions signify areas
where no data are available, and these gaps originate from various sources. Instrument limitations contribute significantly to
these data voids, as sensor sensitivity varies, especially under extreme environmental conditions, and resolution discrepancies
might neglect small-scale features. Additionally, gaps in the satellite's swath coverage during its orbital pass can result in areas
with missing data. Coastal zones are particularly challenging due to the abrupt land-water transition, which complicates the
sensor's ability to distinguish signals accurately. The distinct reflection and refraction characteristics of varied surfaces along
coastlines further complicate data retrieval processes in these regions. The data processing stage also plays a role, as it often
involves rigorous quality control measures that might eliminate data points deemed unreliable or erroneous. Environmental
factors, like areas with high IWV values and extreme meteorological conditions, may exceed the sensor's measurement
capacity, leading to the absence of data in these sections. Moreover, there are also small spots with no data for reasons that
might be transient and are not immediately identifiable, potentially arising from minor satellite function irregularities, data
transmission glitches, or data processing errors.

On October 11, 2009, the AR had been observed in its initial formation stage, extending toward the western part of Ireland.
The structure and alignment of the AR had signalled a nascent phase, with moisture beginning to concentrate along its path.
The following day, the AR exhibited a significant northward extension, reaching beyond 60° N latitudes. This expansion
reflected an intensification of the AR, with moisture transport extending into higher latitudinal regions. On October 13, 2009,
the AR made landfall, impacting the east and southeast coasts of Iceland, and northern Scotland and Ireland. The moisture
content within the AR had reached its peak. The AR's spatial distribution and moisture intensity on that day highlighted the
event's maturity. On the last day, the AR had begun to weaken but had maintained its position over the previously affected
areas. The reduction in moisture content and intensity had signalled the AR's dissipation stage, marking the beginning of the
event's decline. A noteworthy aspect of this AR event had been the transportation of high moisture content to latitudes beyond
60° N, a very uncommon phenomenon – in particular at this time of the year (Parracho et al., 2018). For example, SSMI/S
IWV values had been observed to be as high as 38.7 kg/m² at latitude 60.125° and longitude -12.875° or 36.0 kg/m² at latitude
62.125° and longitude -12.125°, highlighting the significant moisture content transported during this event.





**Figure 2**. Temporal evolution of the AR event over Iceland-UK from October 11 to October 14, 2009, as observed by SSMI/S IWV measurements. The figure illustrates the spatial distribution of the IWV values and includes RO data points.



## 3.2 Calculation Methodology for Specific Humidity and IWV Using RO Data

In this study, we evaluated specific humidity profiles and IWV values. Other parameters, such as relative humidity, mixing ratio, or precipitable water vapor, can be directly calculated using the moist air gas constant ratio parameter (Stull, 2017). As

these other parameters' characteristics are similar to the two parameters, they are not discussed here.

For the WEGC data, specific humidity profiles are already calculated and available in the dataset. For the CDAAC data, the specific humidity $q$, in g/kg, needs to be calculated using the retrieved water vapor pressure $e$, in hPa, and air pressure $p$, also in hPa, profiles. Here, $e$ represents the partial pressure exerted by the water vapor in the air. The specific humidity $q$ is then calculated with the constant $\varepsilon = 0.622$, which is the ratio of the molecular mass of water ($\mu_{\text{water vapor}} \approx 18.015$ g/mol) to the

molecular mass of dry air ($\mu_{\text{dry air}} \approx 28.97$ g/mol) ($\varepsilon = \frac{\mu_{water\,vapor}}{\mu_{dry\,air}}$). This ratio facilitates the conversion from water vapor pressure to specific humidity using the formula:

$$q = \frac{\varepsilon e}{p - (1 - \varepsilon e)} \qquad (2)$$

In this formula, $q$ is the specific humidity, $e$ is the water vapor pressure, and $p$ is the total air pressure.

Second, to calculate the IWV value (kg/m$^2$) of the GNSS-RO profile, the specific humidity of each height from the surface to the maximum retrieved height is integrated, $p_s$ is the surface air pressure, and $g = 9.81$ m/s$^2$ is the mean acceleration of gravity

at sea level at 45° latitude.

$$IWV = -\frac{1}{g} \int_{p_s}^{p} q \, dp \qquad (3)$$

By employing this methodology, we can quantitatively assess the specific humidity profiles and IWV values from the WEGC and CDAAC RO data during AR events.

## 3.3 q-RAER Metric: An Overview and Formula

The Retrieval to A-priori Error Ratio (RAER) profile is a critical tool employed to assess the reliability of retrieved data, offering insights into the relative influence of observed information and background (a priori) information. The RAER profile is expressed as a percentage and is calculated during the optimal estimation process.

It is calculated using the formula:

$$RAER = \frac{\sigma_{ret}}{\sigma_{bg}} \qquad (4)$$

Where $\sigma_{ret}$ contains the square root of the diagonal elements of the retrieval error R, calculated as:

$$R = (\mathbf{B}^{-1} + \mathbf{O}^{-1})^{-1} \qquad (5)$$





And $\sigma_{bg}$ represents the standard deviation of the background error.

In the context of this study, the q-RAER profiles are provided within the WEGC dataset. Specifically, the q-RAER profile
offers a critical evaluation metric for $q$ retrievals, enabling researchers to discern the relative influence and dominance of observed versus background information at various atmospheric layers.

For instance, when assessing the RAER profile in the context of specific humidity retrievals, a notable variance in the dominance of background versus observed information is discerned at different altitudinal levels. The RAER for specific humidity suggests that observed information takes precedence over background information (with RAER values falling below
70 %) at altitudes below approximately 9 km at low latitudes and below 4 km at high latitudes during summer. These observations underline a dynamic interplay between observed and background information, crucial for interpreting the reliability and independence of retrieved atmospheric profiles.

## 4 Results and discussions

### 4.1 Overall comparison between the retrieved specific humidity profiles of CDAAC and WEGC with their background
**profiles**

To assess the impact of background water vapor on the specific humidity profiles retrieved from RO data, a comparison is conducted using the complete dataset of occultations during AR events, along with their corresponding background and reference profiles. In this subsection, we have selected four examples to be discussed here. These examples were chosen to represent a range of behaviours and patterns observed in the data, and they provide a concise yet comprehensive overview of
the key findings. The selected profiles are those located inside or in the vicinity of the AR over the oceanic area. Among these, the first example serves as a general case, showcasing a common pattern observed across various profiles. The remaining three examples are selected to highlight instances where the background profile and the 1D-Var profile exhibit different and intriguing patterns. These examples are not exhaustive but provide insights into the diverse behaviours and characteristics observed in the dataset.

### 4.1.1 Iceland-UK 2009 AR event

Figure 3(a and b) illustrates the specific humidity profiles for the Iceland-UK 2009 AR event observed by a satellite of the COSMIC-1 constellation, including CDAAC 1D-Var, WEGC 1D-Var, ERA, GFS, ECMWF-r and ECMWF-b, as well as the q-RAER profile for the WEGC retrieved $q$ profile. The distribution of IWV detected by the SSMI/S F17 satellite, along with the location of the RO event, is depicted in Figure 3(c). Figure 3(e) further displays the RO tangent point trajectories and
reference points for both centres, with the lowest 14 km of each centre's RO event indicated by a black dashed line and the highest altitude of the RO marked by a black triangle. The highest retrieved altitudes for WEGC and CDAAC were 80 km and 40 km, respectively. This delineation of the RO event's lower tropospheric region is essential because of the significant



humidity gradient near the AR. In this example, the RO event is situated within the AR, where the SSMI/S IWV is approximately 30 kg/m², as shown in Figure 3(c). All six specific humidity profiles generally exhibit similar patterns, despite variations in their IWV values. These IWV values, while slightly different, are in relatively proximity to the SSMI/S IWV.

**Figure 3.** (a) Retrieved specific humidity profiles of CDAAC and WEGC with their reference and background profiles for a satellite of the COSMIC-1 constellation during the Iceland-UK AR event on 13 October 2009. All IWV values in panel (a) are in kg/m². The WEGC specific humidity RAER profile is shown in plots (b). (c) Distribution of the IWV detected by the SSMI/S F17 satellite with the location of the RO events. (d) RO tangent point trajectory paths of WEGC and CDAAC.





The retrieved CDAAC profile displays more fluctuation compared to other profiles. Similar fluctuations in the CDAAC $q$

profile are also observed in other examples. A possible explanation for this behaviour could be a lower vertical correlation in the error covariance matrices. Note that the SSMI/S IWV map does not necessarily represent the exact IWV distribution at the time of the GNSS-RO event, since these maps are only available once per day.

### 4.1.2 UK 2020 AR event

Figure 4(a) illustrates specific humidity profiles for the 2020 AR event observed by MetOp-A. In this example, the RO event

is located in the northern region of the UK 2020 AR event, near its edge.

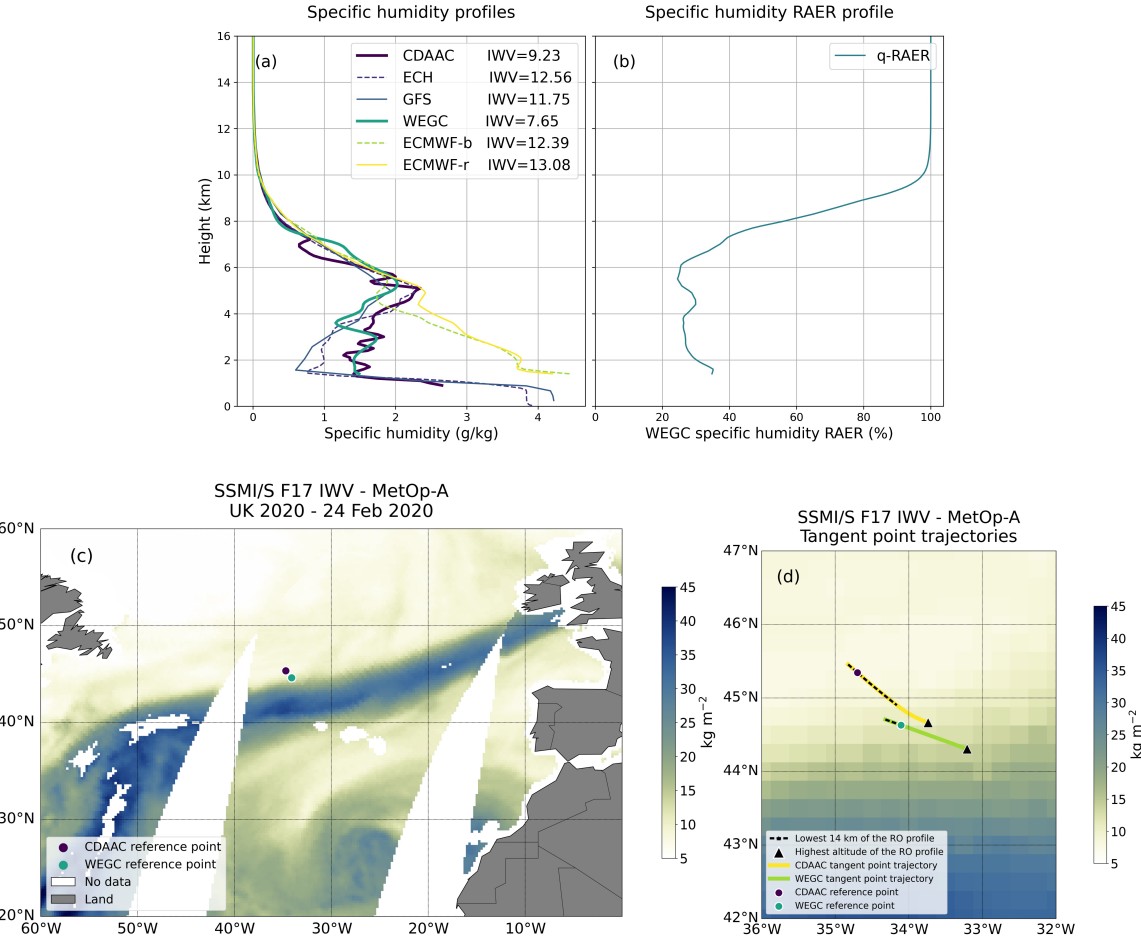

**Figure 4. (a)** Retrieved specific humidity profiles of CDAAC and WEGC with their reference and background profiles for the MetOp-A satellite during the UK AR event on February 24, 2020. The WEGC specific humidity RAER profile is shown in plot **(b)**. **(c)** Map of the

IWV detected by the SSMI/S F17 satellite with the location of the RO events. **(d)** RO tangent point trajectory paths of WEGC and CDAAC.





While the results in 4.1.1 are typical for profiles in the core region of the AR, we chose this GNSS-RO profile, since it nicely illustrates, how the humidity retrievals operate, when the background profiles are very different – because they have been extracted at different locations due to the different interpretation of the occultation location. Above 7 km, all specific humidity profiles still exhibit similar pattern. For altitude above ~ 9 km it is not surprising that the WEGC profiles follows closely the ECMWF-b profile, since the RAER values are close to 100% and the retrieved profiles is therefore heavily background-dominated. However, below ~ 7 km, where the RAER falls below 40% and the observed data gain a high weight, there are large differences between retrieved and background profiles, in particular below ~5 km, where the retrieved profile is way drier than the background profile. For the CDAAC retrieval the situation is different, here the background profile has been extracted in a very dry area (ECH stands for ECMWF profiles with high vertical resolution). For the CDAAC data, we don't have a value similar to RAER, but the large difference between retrieval and background suggests, that there is also a hight weighting of the measurements in this altitude range. Interestingly, the WEGC and CDAAC retrievals agree quite well – even though the background data are so different. This increases confidence in the retrieved data, and suggests that the GNSS-RO data provide indeed valuable humidity profile information – beyond the background.

Note that both retrieved profiles don't reach the lowest km in this case (WEGC background and reference profiles are only stored in the altitude range, where there is also a retrieved profile), the computed IWV values therefore underestimate the true values.

### 4.1.3 California 2019 AR event

Figure 5 illustrates an example of an AR event in California in 2019. The selected RO event in this case is located near the AR event, south of Hawaii (represented in the map as a grey triangle), as shown in Figure 5(c). Between 6 km and 2 km, the ECMWF-b profile indicates higher humidity than the ERA5 and WEGC profiles. The WEGC q retrieval profile within this altitude range is significantly drier than its background profile. A moderate difference is observed between the CDAAC and ERA5 retrieved q profiles in the layer from 4 km to 2 km. The WEGC q profile stops at an altitude of 500 m, whereas the CDAAC profile extends to the lowest altitude. Although the CDAAC profile shows less humidity below 5.5 km, the calculated IWV value for CDAAC is slightly higher (WEGC IWV = 18.33 kg/m$^2$ and CDAAC IWV = 20.75 kg/m$^2$) because the WEGC q profile lacks humidity information in the lowest 500 m. Both background profiles have higher IWV differences of approximately 7.67 and 6.99 kg/m$^2$ (ERA5 and ECMWF-b respectively) compared with their respective retrieval q profiles, since the lowest part of the actual GNSS-RO profile extends into much drier areas than the vertical background profiles (Note that the CDAAC algorithm better represents the actual TP trajectory).

The q-RAER profile also indicates that below 6 km, the observation data significantly influence the retrieved q profiles. The WEGC retrieval q profile is drier than its background profile between 5 and 2 km, whereas the CDAAC profile is wetter than its background profile. As in section 4.1.2 it seems that both retrievals try to bring the retrieved profiles into close agreement-even though the background profiles are very different in this altitude range.





Despite methodological differences, there is a remarkable consistency in the final q profiles across the centres. This
alignment, while exhibiting some variations, underscores the resilience and adaptability of the retrieval processes across
different methodologies. Notably, these variations in the q profiles can be attributed to the differences in background water
vapor information used in the 1-DVAR methods, as exemplified in Figure 6. This highlights the nuanced interplay between
methodological choices in data retrieval and the influence of background atmospheric conditions.

**Figure 5. (a)** Retrieved specific humidity profiles of CDAAC and WEGC with their reference and background profiles for the MetOp-A
satellite during the California 2019 AR event. The WEGC specific humidity RAER profile is shown in plot **(b)**. **(c)** Distribution of the IWV
detected by the SSMI/S F17 satellite with the location of the RO events. **(d)** RO tangent point trajectory paths of WEGC and CDAAC.





### 4.1.4 Southern Africa 2009

Figure 6 depicts the Southern Africa 2009 AR event, by examining a profile from the COSMIC-1 constellation. The RO event is located on the southern edge of the AR and probes the atmosphere downward, moving toward the outside of the AR. In this example, the ECMWF-b profile (taken closer to the AR) displays significantly higher q values than WEGC between altitudes

of 3.8 km and 2 km, where the observation data are dominant (q-RAER profile). Moderate differences are also observed below 2 km and above 4.3 km, while between 3.8 km and 4.3 km and up to 5.8 km, the ECMWF- b profile is drier than the WEGC profile. Between 2 km and 3.8 km, the CDAAC profile is also drier than its prior information.

Figure 6. (a) Retrieved specific humidity profiles of CDAAC and WEGC with their reference and background profiles for a COSMIC-1 satellite during the Southern Africa AR event 2009. The WEGC specific humidity RAER profile is shown in plot (b). (c) Distribution of the IWV detected by the SSMI/S F17 satellite with the location of the RO event. (d) RO tangent point trajectory paths of WEGC and CDAAC.






The CDAAC profile stops at 1 km, while the WEGC profile reaches the lowest 200 m altitude. Consequently, the IWV value of CDAAC is lower than the others. The IWV value of ECMWF-b is closer to the observed SSMI/S IWV, which is about 17 kg/m$^2$, and the WEGC IWV is 6.25 kg/m$^2$ less than the SSMI/S observation. This pattern is also observed for the CDAAC IWV value, which is about 6.42 kg/m$^2$ drier than its background ERA IWV. Again, we note that the SSMI/S IWV map does not necessarily represent the exact IWV distribution at the time of the GNSS-RO event.

It seems that the dry layer observed by both retrievals between ~2 km and ~3 km may in fact have been even drier. It is better represented by CDAAC, where the background was already quite dry, whereas the WEGC retrieval could not reach such low values, since it was somewhat confined by the background profile, which was much too humid.

When analysing specific humidity profiles retrieved from CDAAC and WEGC and comparing them to background profiles within AR events, noticeable variations and patterns emerge. The precise location of tangent point trajectories and reference
points is crucial, especially when they are situated at the edges of AR events, where the humidity gradient is often significant, since background data are extracted at very different locations with different humidity regimes. This variance in the location of tangent point trajectories between CDAAC and WEGC adds another layer of complexity to interpreting the data. Understanding and accounting for these subtle differences in trajectory locations within a single RO event is essential for a more accurate analysis and interpretation of the humidity profiles during AR events, enhancing the reliability and precision of
the findings derived from such data.

**4.2 IWV comparison between CDAAC and WEGC**

In this subsection, we compare the IWV data retrieved from the two centres, CDAAC, and WEGC. RO data were collected from all active GNSS-RO satellites during the investigated AR events within the entire study domain (see Fig. 1 and Table 1), ensuring that we capture a wide range of IWV values retrieved by RO. To maintain consistency in our comparison, we included
only data from satellites for which RO retrieval data were available in both the WEGC and CDAAC databases. From the six AR events studied, we have chosen three with the most interesting results for discussion in this paper. This focused approach allows for a clearer, more engaging analysis without overwhelming the reader with too much information.

We specifically calculate the IWV for those profiles that reach altitudes of 1 km, 2 km, 500 m, and 200 m in both centres. In other words, we consider only those RO profiles from both centres that extend to these specified altitudes for our IWV
calculations.

We chose these specific altitudes for two primary reasons. First, lower altitudes tend to have higher concentrations of water vapor, and calculating the IWV at these levels provides a more representative assessment of the water vapor distribution within the atmosphere. Second, not all RO profiles are capable of reaching the lowest altitudes, so categorizing the data into these four different altitudes allowed us to capture a more comprehensive range of RO profiles, ensuring a more robust dataset for
our analysis. Subsequently, the IWV values were integrated from these defined altitudes up to the highest retrieved RO profile altitude, facilitating a comprehensive comparison of IWV values across the selected altitudes.





### 4.2.1 UK 2020 AR event

Analysis of the UK 2020 AR event (see section 4.1.2) reveals that the IWV measurements from the WEGC and CDAAC
exhibit slight differnces (Fig. 7), incorporating those points:

163 GNSS-RO profiles reached at least 2 km altitude in both datasets, where CDAAC data show slightly smaller values than
WEGC with a mean bias error (MBE) of -0.22 kg/m$^2$ and a root mean square error (RMSE) of 1 kg/m$^2$. The relationship is
described by the regression line equation (y = 0.97x - 0.1), indicating a nearly one-to-one relationship with a minor offset.

As we descend to 1 km, the number of simultaneous observations decreases to 134, since fewer profiles reach down to this
altitude. The MBE increased to -1.01 kg/m$^2$ and the RMSE to 2.25 kg/m$^2$. The correlation also deviated slightly, represented
by the equation (y = 0.87x). On average, the CDAAC values are around 87 % of the WEGC values. A closer inspection,
however, reveals that this systematic difference is largely due data from the COSMIC-2 constellation.

Further down at 500 m, the observations decreased even more to 103. The MBE was -0.83 kg/m$^2$ and the RMSE was slightly
higher 2.5 kg/m$^2$, and CDAAC COSMIC-2 values are again systematically lower.

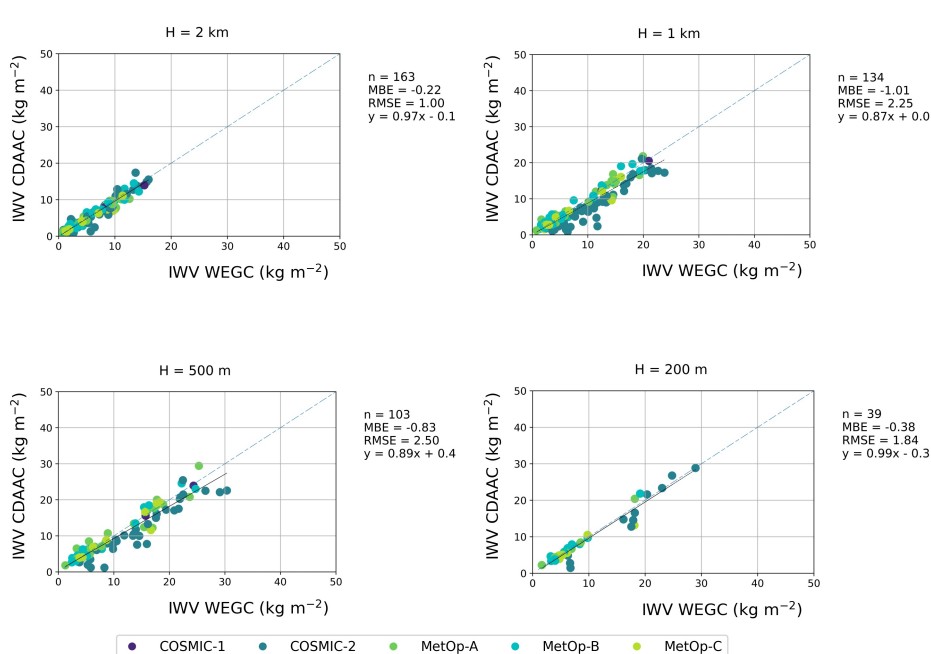

**Figure 7.** Comparison of IWV (kg/m$^2$) values derived from GNSS-RO profiles reaching specified minimum altitudes during the UK 2020
AR event. Each subplot contrasts IWV values from WEGC with those from CDAAC. The four panels correspond to RO profiles reaching
down to 2 km, 1 km, 500 m, and 200 m respectively. RMSE and MBE units are in kg/m² as well.



Lastly, only 39 profiles in both datasets reach down to the lowest altitude of 200 m, but the correlation is again very close to a one-to-one relationship (y = 0.99x - 0.3). The MBE is -0.38 kg/m$^2$, the RMSE is 1.84 kg/m$^2$.

Predictably, as we descend in altitude, the IWV values generally increase, due to the higher humidity levels commonly found at lower altitudes. This is, however, not necessarily and always the case, since that there is also a tendency that GNSS-RO

profiles reach further down in dry areas. A small ensemble of profiles reaching very low altitudes could therefore oversample particularly dry regions (see the example in below in section 4.2.2).

In conclusion, for the UK 2020 AR event, the IWV data from both CDAAC and WEGC are closely aligned, except for the COSMIC-2 profiles reaching down to 1 km and 500 m respectively. The reason for this is currently unknown and justifies further analysis.


### 4.2.2 Southern Africa 2009 AR event

The IWV measurements from the WEGC and CDAAC are compared in Figure 8 for Southern Africa 2009 (see section 4.1.4) for different minimum altitudes.

Based on 174 GNSS-RO profiles, which at least reached the highest altitude of 2 km, the IWV data from the two centres

display a strong correlation (y = 0.95x + 0.3), with an MBE of just -0.01 kg/m$^2$ and an RMSE of 0.8 kg/m$^2$.

Descending to 1 km, the number of simultaneous observations reduces to 139. The correlation remained strong, though the offset is higher (y = 0.95x + 1.0). The MBE increases slightly to 0.48 kg/m$^2$, and the RMSE to 1.27 kg/m$^2$.

Further down at 500 m, there is a near one-to-one correlation, but with an even more pronounced offset (y = 0.97x + 1.8), resulting in an increased MBE of 1.34 kg/m$^2$ and am RMSE of 1.97 kg/m$^2$.

At the minimum altitude of 200 m, observations dropped significantly to just 11. Over recent year, GNSS-RO profiles reached lower and lower altitudes, but back in 2009 the lowest 200 m have still been out of reach for most profiles.

Due to the limited ensemble size, these results may be less stable Here, the correlation between the two datasets shows a greater deviation with a substantial positive offset (y = 0.89x + 4.6). The MBE is notably higher with 2.48 kg/m$^2$, and the RMSE reaches 3.2 kg/m$^2$.

In conclusion, for the Southern Africa 2009 AR event, the IWV data from CDAAC and WEGC exhibit a strong correlation at higher altitudes, with some deviations becoming apparent as we descend. Overall, while the two datasets offer reliable insights across different altitudes, they also bring forth the importance of understanding the subtleties and potential discrepancies inherent in the data, especially at lower altitudes. Interestingly, while the maximum IWV recorded from an RO event at 500 m reached 37 kg/m², it's worth noting that the maximum IWV for the RO profiles descending further to 200 m was 33 kg/m², due

to selective sampling of drier regions (see section 4.2.1 above). This observation highlights that not all RO profiles reaching the lowest altitudes include the highest IWV values.



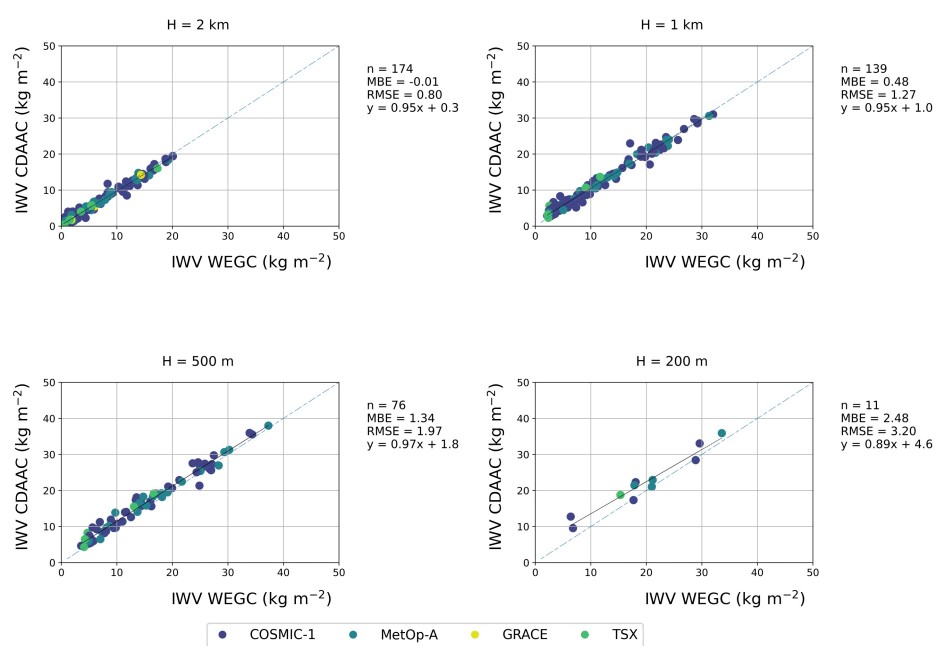

**Figure 8.** Comparison of IWV values (in kg/m²) derived from GNSS-RO profiles reaching specified minimum altitudes during the Southern Africa 2009 AR event. Each subplot contrasts IWV values from WEGC with those from CDAAC. The four panels correspond to RO profiles reaching down to 2 km, 1 km, 500 m, and 200 m, respectively. RMSE and MBE are given in kg/m².

### 4.2.3 California 2017 AR event

The IWV comparison for the California 2017 case highlights a generally close agreement between the WEGC and CDAAC at all altitudes, with slight variations in the degree of offset (see Fig. 9).

For the RO profiles reaching to at least 2 km, with a sample size of 111 observations, the IWV data correlation is very strong, (y = 0.97x + 0.1), having an MBE of only -0.09 kg/m² and an RMSE of 0.68 kg/m². Descending to 1 km, the number of observations moderately decreases to 95. The high correlation persists (y = 0.98x + 0.3) with a marginally increased positive offset, with an MBE of 0.15 kg/m² and an RMSE of 1.09 kg/m².

Descending to 500 m, based on 81 observations, the correlation between the two centres' IWV data remained extremely consistent (y = 1.00x + 0.5). The MBE increased to 0.45 kg/m², and the RMSE to 1.41 kg/m².

Lastly, further down at 200 m altitude, observations are based on 34 data points. The correlation remained surprisingly reliable with a one-to-one correlation and a minimal positive offset (y = 1.00x + 0.1), with an MBE of only 0.16 kg/m² and an RMSE of 1.57 kg/m².





In the California 2017 AR event, the IWV data from WEGC and CDAAC show very close agreement at all altitudes, with only slight variations noted. The analysis, conducted at various altitudes and based on different numbers of observations, consistently yields strong correlations between the two data sets.

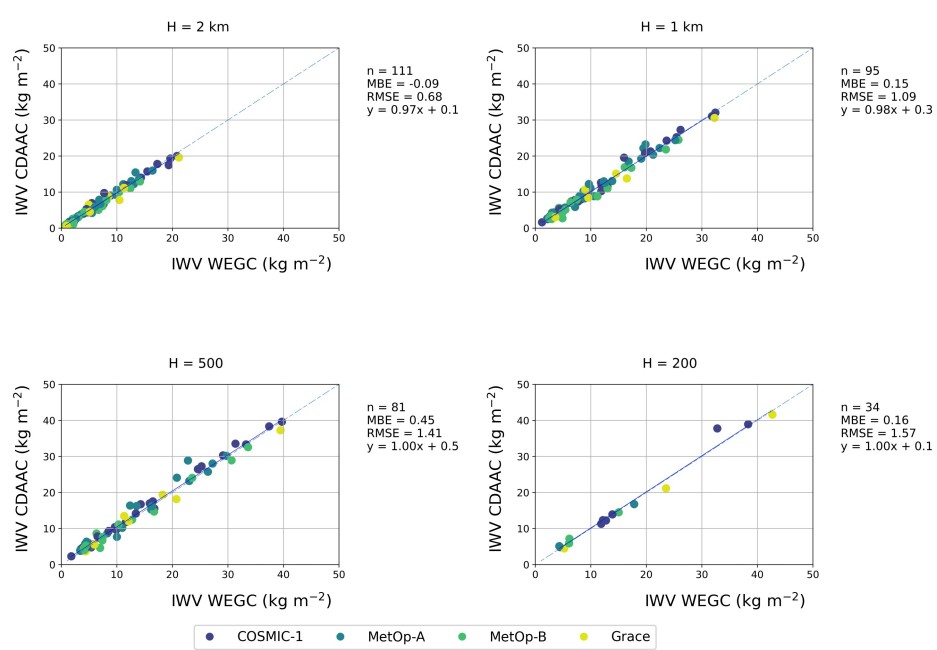

**Figure 9.** Comparison of IWV values derived from GNSS-RO profiles reaching specified minimum altitudes during the California 2017 AR event. Each subplot contrasts IWV values from WEGC with those from CDAAC. The four panels correspond to RO profiles reaching down to 2 km, 1 km, 500 m, and 200 m. RMSE and MBE units are in kg/m².

In examining the IWV data from the UK 2020, Southern Africa 2009, and California 2017 AR events, the correlation between CDAAC and WEGC is generally strong. In summary, across all three events and various altitudes, the data demonstrate a consistently strong correlation, with MBEs ranging approximately from -1.01 to 2.48 kg/m² and RMSEs from 0.68 to 3.2 kg/m².

## 4.3 GNSS-RO and SSMI/S IWV comparison

This section focuses on a comparative analysis of IWV values derived from two distinct observational sources. The first set of IWV values is obtained from RO profiles reaching down to an altitude of 200 m and are sourced from CDAAC and WEGC. This calculation of IWV from the lowest 200 m to the uppermost altitude of the RO profiles is designed to capture all possible



humidity content, given that not all RO profiles can reach the Earth's surface. The second set of IWV values comes from observations made by SSMI/S.

The comparison is structured to assess how well these two data sources align or differ in terms of IWV measurements. In this analysis, a critical aspect to consider is the spatial resolution of the SSMI/S measurements, which is approximately 0.25° × 0.25° in latitude and longitude, translating to a spatial coverage of about 27 km for each grid square at the equator. Furthermore, the obliquity of the RO profiles is an important factor in this methodological consideration. It is essential to understand that RO profiles are not perfectly vertical (see section 2.1.1), which influences the representation of the atmospheric

column in comparison to the SSMI/S observations.

For the purpose of comparison with the SSMI/S IWV values, it is crucial to accurately determine the most representative latitude and longitude coordinates. Given the spatial resolution of the SSMI/S data, the most representative coordinates for this comparison are identified based on the lowest 2 km segment of the RO profiles. This decision is informed by the fact that each SSMI/S grid, covering approximately 27 km, can encompass the lowest 2 km of the RO data. In most cases, the lowest 2 km

of the RO data are contained within one of the SSMI/S grids. By selecting the latitude and longitude from this part of the RO profile, the study ensures a more accurate and meaningful comparison of the IWV values between the two observational methods

### 4.3.1 Iceland-UK 2009 AR event

In Figure 10(a), based on 111 observations, the correlation yields an MBE of -3.35 kg/m$^2$ and RMSE of 6.41 kg/m$^2$. This

relationship between the CDAAC and SSMI/S datasets is encapsulated by the equation y = 0.79x + 1.7. This suggests that the CDAAC values approximate to about 79 % of the SSMI/S values, accompanied by a noticeable offset.

On the other hand, Figure 10(b) contrasts the WEGC IWV against the SSMI/S IWV with 96 observations. The correlation here exhibits an MBE of -3.20 kg/m$^2$ and an RMSE of 5.92 kg/m$^2$. The relationship can be characterized by y = 0.80x + 0.7, implying that WEGC values are closely aligned, being around 80 % of the SSMI/S values, and with a slightly diminished

offset compared to the CDAAC data. Both CDAAC and WEGC IWV datasets, as illustrated in Figure 10, exhibit a reasonably close alignment with the SSMI/S IWV values, but both underestimate the SSMI/S IWV values in a similar way.

Notably, in Figure 10(b), the WEGC dataset is slightly smaller, with 15 fewer events, than the CDAAC dataset in this case study. Additionally, the number of RO profiles by the GRACE satellite is 6 in 10(a) and only one in 10(b). Furthermore, the number of COSMIC-1 observations is considerably larger in 10(b), while Metop-A observations are more prominent in 10(a).

Since the water vapor content in the lowest 200 m is missing in the GNSS-RO-derived IWV values we have to expect a (slight) underestimation, but the results are somewhat lower than anticipated.





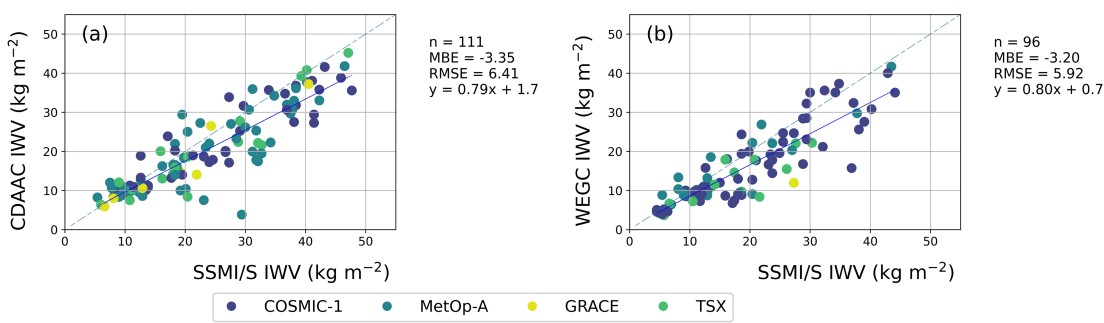

**Figure 10.** Comparison of IWV values (in kg/m²) derived from GNSS-RO profiles from CDAAC (**a**) and WEGC (**b**) and SSMI/S observations during the Iceland-UK 2009 AR event. The IWV values from each GNSS-RO profile are computed from 200 m altitude up to the uppermost limit of each profile. The SSMI/S IWV values are extracted at the latitude and longitude that corresponds to the 2 km altitude of the GNSS- RO profiles. MBE and RMSE values are given in kg/m².

### 4.3.2 Southern Africa 2009 AR event

The second comparison focuses on the Southern Africa AR event in 2009 as visualized in Fig. 11. Delving into Fig.11(a), based on 39 observations, there emerges a correlation with a marked negative offset of the CDAAC data (y = 0.92x – 3.7), leading to an MBE of -6.04 kg/m² and an RMSE of 7.2 kg/m². As a result, almost all CDAAC values are lower than the corresponding SSMI/S values.

Transitioning to Figure 11(b), which contrasts WEGC IWV against SSMI/S IWV, the dataset is drawn from 67 observations. The correlation manifests an MBE of -4.78 kg/m² and an RMSE of 6.71 kg/m². This relationship can be expressed by the equation y = 0.71x + 1.6, indicating that WEGC around 10 kg/m² are quite well represented, while higher values are underrepresented.

In this AR event, the CDAAC dataset includes 39 events, whereas WEGC includes 67 events, whereas in most other cases there are more CDAAC than WEGC events reaching down to 200 m altitude. Notably, there are no TSX and GRACE RO events observed to reach the lowest 200 m altitude in both Figure 11(a) and 11(b).





Southern Africa 2009, COSMIC-1, MetOp-A, GRACE, TSX

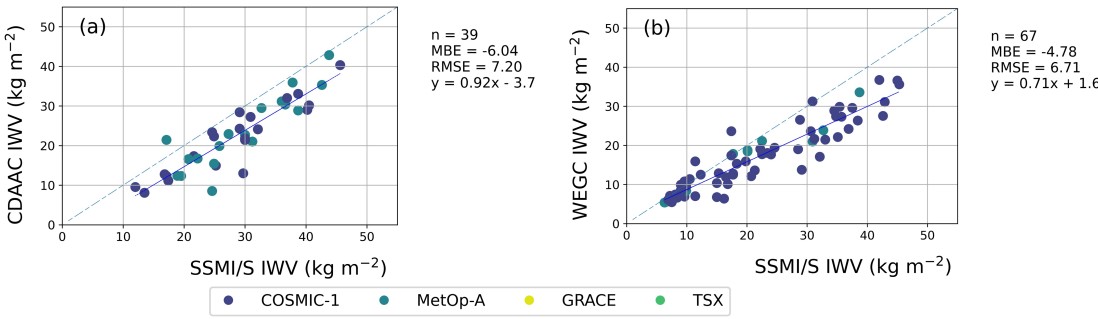

**Figure 11.** Comparison of IWV values (in kg/m²) derived from GNSS-RO profiles from CDAAC (**a**) and WEGC (**b**) and SSMI/S observations during the Southern Africa 2009 AR event. The IWV values from each GNSS-RO profile are computed from 200 m altitude up to the uppermost limit of each profile. The SSMI/S IWV values are extracted at the latitude and longitude that corresponds to the 2 km altitude of the GNSS- RO profiles. MBE and RMSE values are given in kg/m².

### 4.3.3 California 2017 AR event

The final analysis focused on the California 2017 AR event. The outcomes of this assessment are visualized in Figure 12. Starting with Figure 12(a), from a sample of 126 observations, the correlation yielded an MBE of -2.28 kg/m² and an RMSE of 4.92 kg/m². The relationship between the CDAAC and SSMI/S datasets can be expressed by the equation $y = 0.82x - 1.1$.

Proceeding to Figure 12(b), the dataset comprises only 40 observations reaching down to 200 m. Here, the correlation manifests an MBE of -1.06 kg/m² and an RMSE of 4.47 kg/m². This relationship is characterized by the equation $y = 0.83x + 1.3$, signifying that the WEGC values are close to 83 % of the SSMI/S values, but with a contrasting positive offset.

Especially during this event, the numbers of ensemble members in both datasets are strikingly different, which is remarkable since both datasets are based on the same low-level data.





California 2017, COSMIC-1, MetOp-A, MetOp-B ,GRACE

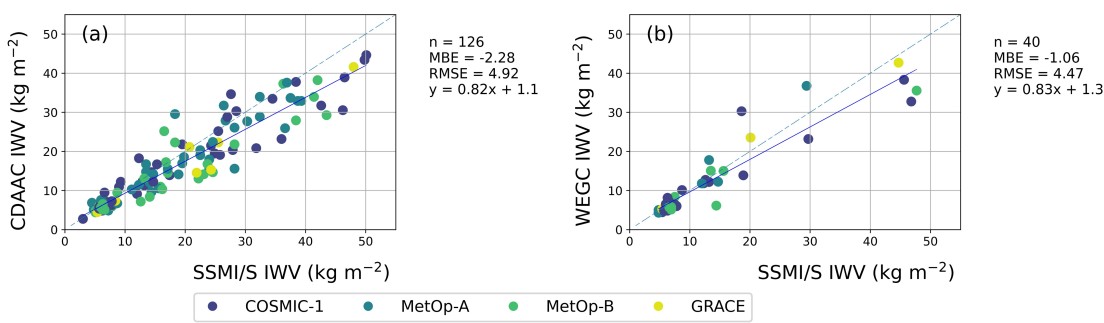

**Figure 12.** Comparison of IWV values (in kg/m²) derived from GNSS-RO profiles from CDAAC (**a**) and WEGC (**b**) and SSMI/S observations during the California 2017 AR event. The IWV values from each GNSS-RO profile are computed from 200 m altitude up to the uppermost limit of each profile. The SSMI/S IWV values are extracted at the latitude and longitude that corresponds to the 2 km altitude of the GNSS- RO profiles. MBE and RMSE values are given in kg/m².

## 4.4 Comparison of CDAAC and WEGC IWV values with background profiles

In this subsection, we examine the IWV values obtained from CDAAC and WEGC centres in relation to their corresponding background profiles. The comparison has been conducted for those profiles which reach at least to altitudes of: 2 km, 1 km, 500 m, and 200 m. By considering the background profiles, we aimed to assess the differences between the background and retrieved IWV values of the two centres. Through this investigation, we seek to identify any discrepancies or biases between the IWV values and the corresponding background profiles.

### 4.4.1 Iceland-UK 2009 AR event

For the Iceland-UK 2009 AR event, this resulted in a set of eight plots, with four comparing CDAAC IWV against ERA IWV shown in Figures 13(a-d) and another four contrasting WEGC IWV against ECMWF-b as seen in Figures 13(e-h).

For the CDAAC IWV versus ERA IWV comparison, a general trend can be observed. As altitude decreases from 2 km to 200 m (Figures 13(a-d)), the MBE grows more negative, indicating a progressively larger underestimation of the CDAAC IWV values compared to the ERA background. Concurrently, the RMSE also escalated, suggesting increasing variability in the comparison. The regression equations showed a steady decrease in the slope, implying a less direct correlation between the datasets as altitude decreased. The most pronounced deviation was noticed at 200 m altitude, where the equation is $y = 0.66x + 2.4$, denoting a significant positive offset and reduced slope.



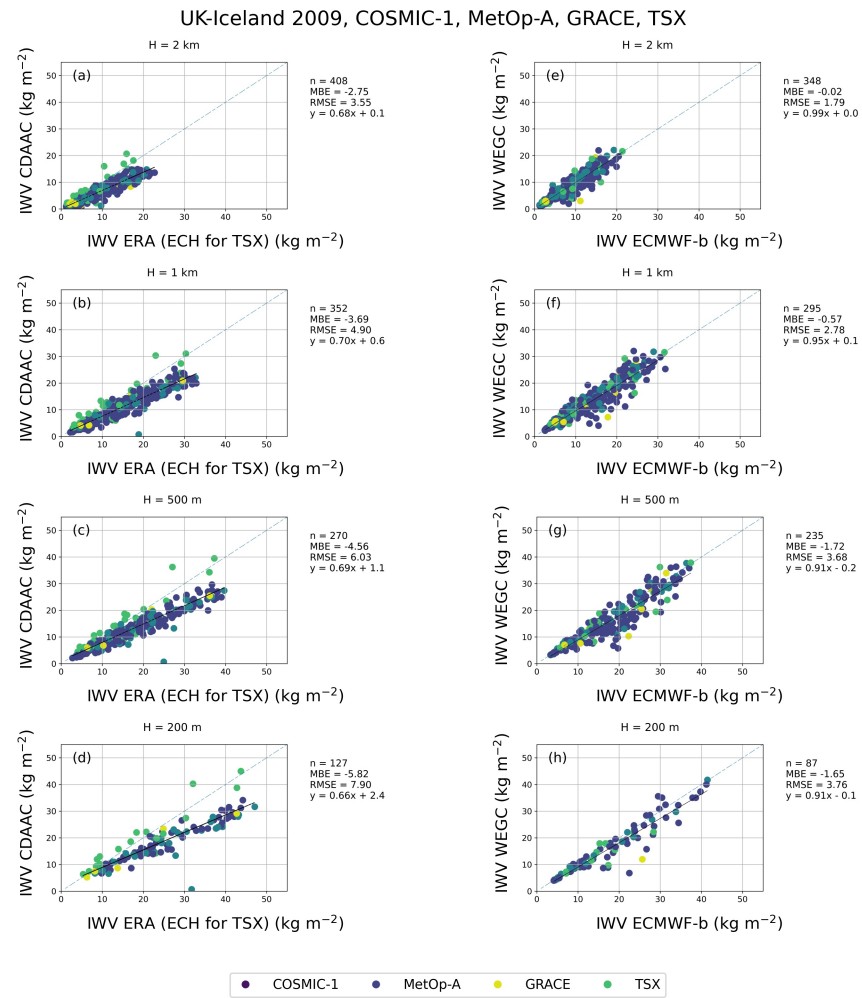

**Figure 13.** Comparison of IWV values (kg/m²) retrieved from CDAAC and WEGC centres against their respective background profiles during the Iceland-UK 2009 AR event. Eight subplots are distributed over two columns, examining four distinct altitudes: 2 km, 1 km, 500 m, and 200 m. In the left column, plots **(a)**, **(b)**, **(c)**, and **(d)** display the relationship between IWV values from ERA background profiles and IWV values from CDAAC, noting that the TSX satellite uses ECH as the background information. Conversely, in the right column, plots **(e)**, **(f)**, **(g)**, and **(h)** present comparisons between IWV values from ECMWF-b background profiles and IWV values from WEGC. RMSE and MBE units are in kg/m².

On the other hand, the WEGC versus ECMWF-b comparison shows a different picture. Throughout the altitudinal range of 2 km to 200 m (Figures 13(e-h)), the MBE values remain less than 2 kg/m², suggesting a closer alignment of WEGC values to the ECMWF-b background. The RMSE values, while increasing as altitude decreases, are generally lower than their CDAAC





counterparts. The regression equations exhibit a consistent and high correlation between the datasets across all altitudes, with slopes nearing unity at higher altitudes and a minimal offset.

In the in-depth analysis of the IWV values against their corresponding background profiles, an important distinction emerges regarding the CDAAC background data sources. While most satellites employ ERA as their background data, TSX uniquely utilizes ECH profiles. This distinction becomes particularly evident when observing Figures 13(a-d).

From Figures 13(a-d), a notable pattern arises: the TSX data points predominantly align with the 1:1 line, indicating a direct correlation with the background data. In stark contrast, data from other satellites consistently tends to underestimate the IWV values. This implies that, among the satellites examined, TSX exhibits a closer alignment and is more correlated with the background data than other satellites.

The presence of TSX data along the 1:1 line reinforces its accuracy and alignment with the ECH profiles it uses as a background. This emphasizes the importance of the choice of background profiles and how they can influence the retrieved IWV values. Noteworthy, while background profiles generally exhibit higher IWV values in figures 11(a-d), the retrieved IWV values in CDAAC tend to be drier, with the exception of TSX data.

A close agreement between retrieved and background-derived IWV is not necessarily good or bad, even more since we know from section 4.3.2 that there is a close agreement of the IWV values derived from CDAAC and WEGC data, respectively. A quite possible explanation is that the ERA-derived IWV values are indeed too high, whereas the CDAAC retrieval brings the profiles into closer agreement with the "truth" and (in this case) with the WEGC data.

### 4.4.2 UK 2020 AR event

In the second study case centered on the UK 2020 event, an analysis was undertaken to evaluate the IWV values from the CDAAC and WEGC centres against their respective background profiles across four distinct altitudes: 2 km, 1 km, 500 m, and 200 m. These results are visually represented in Figure 14(a-h).

For the CDAAC IWV, which uses ECH profiles as its background data, the correlation with the background is observed across Figures 14(a-d). At 2 km, the correlation exhibits a slight underestimation with an MBE of -0.4 kg/m². and an RMSE of 1.1 kg/m². As altitude decreases, this underestimation becomes more pronounced, reaching an MBE of -0.78 kg/m². at both 500 m and 200 m. The regression slopes for these altitudes indicate that the CDAAC values are generally around 85% to 91% of the background values, with a slight positive offset.

Figures 14(e-h) showcase the comparison for WEGC IWV data. At 2 km altitude, the data almost mirrors the background with an MBE of 0.01 kg/m². However, as we descend in altitude, there's a gradual shift towards underestimation, reaching an MBE of -1.12 kg/m² at 500 m. The regression slopes remain relatively close to unity, especially at higher altitudes, suggesting a strong alignment with the background data.





Comparing these findings with the previous study case for the UK-Iceland 2009 event, some distinctions emerge. In the 2009

event, while the retreivals for most satellites at CDAAC used ERA as the background, the exception of TSX utilizing ECH profiles stood out due to its close alignment along the 1:1 line. In the 2020 event, the utilization of ECH profiles as the background for CDAAC IWV seems to offer a more consistent and aligned IWV estimation across altitudes compared to when ERA was predominantly used as the background in the 2009 event.

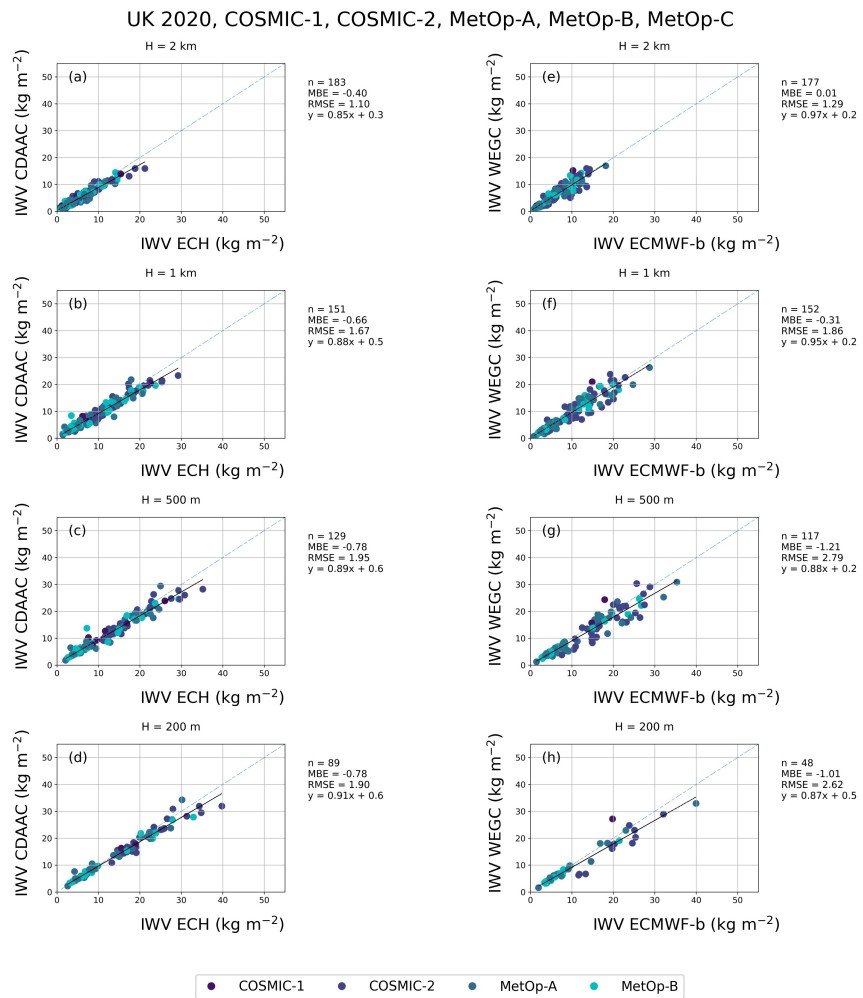

**Figure 14.** Comparison of IWV values retrieved from CDAAC and WEGC centres against their respective background profiles during the UK 2020 AR event. Eight subplots are distributed over two columns, examining four distinct altitudes: 2 km, 1 km, 500 m, and 200 m. In the left column, plots **(a)**, **(b)**, **(c)**, and **(d)** display the relationship between IWV values from ECH background profiles and IWV values from CDAAC. Conversely, in the right column, plots **(e)**, **(f)**, **(g)**, and **(h)** present comparisons between IWV values from ECMWF-b background profiles and IWV values from WEGC. RMSE and MBE units are in kg/m².





The choice of ECH profiles for CDAAC IWV in the 2020 event appears to enhance its correlation, echoing the superior
alignment observed with TSX in the 2009 event, which also used ECH profiles. This emphasizes the important role of
background data in shaping the accuracy and reliability of satellite-derived IWV values, with ECH profiles emerging as a
potentially more aligned choice in these study cases.

**4.5. Comparative Analysis of WEGC specific humidity profiles with ECMWF Background Data Across Events**

In this subsection, we compare the specific humidity profiles retrieved by WEGC with their corresponding ECMWF
background data across different events. The goal of this analysis is to understand the extent and nature of discrepancies, if
they are, between the retrieved and background data, and analyse the effect of background water vapor on the retrieved q.

Initially, the differences between the specific humidity values retrieved by WEGC ($q_{wegc}$) and those from ECMWF ($q_{ECMWF-b}$)
were computed for those profiles reaching a minimum altitude of 500 m. These differences are visualized in a plot against
altitude (Fig. 15), offering a clear representation of how deviations vary with height. To ensure a substantial data sample while
maintaining the ability to analyse specific altitudes like 2 km and 1 km, only profiles reaching down to the lowest 500 m are
incorporated in this comparison. A decision to narrow down to 200 m would have resulted in a restricted sample size,
potentially skewing the insights. Subsequent to this, for the plotted differences, the mean specific humidity and its standard
deviation (std) are computed for each satellite individually.

Furthermore, the relative specific humidity difference (RSHD) was calculated and visualized for each event, providing a
quantitative measure of the accuracy of the retrieved data in relation to the background. Lastly, to assess the variability and
dispersion of the relative humidity differences, the interquartile range (IQR) and standard deviation for the relative specific
humidity differences are plotted. This comprehensive approach ensures a holistic understanding of the alignment between the
WEGC retrieved data and the ECMWF background across different altitudes and satellites.

Upon reviewing Fig. 16 which, in combination, covers six distinct events, several patterns emerge. The plots showcase
specific humidity relative differences when comparing WEGC retrievals against ECMWF background data. This difference
representation, spread across different altitudes, grants a layered perspective into the retrievals for each event.
It is evident that the differences are generally minimal at higher altitudes. This is inferred from the concentration of data
around the zero-difference mark in these regions, signifying that the humidity values are small to begin with, and that the
WEGC retrievals closely mirror the ECMWF background data. However, as one moves to lower altitudes, the spread of
differences broadens, indicating an increase in discrepancies between the retrievals and the background. This is not a bad
thing, since it just means that retrievals differ from their respective backgrounds, which is – in general – related to an
increase in information content.
Another striking observation is the variability between different datasets or satellites within each event. While some datasets
exhibit a tight clustering of differences around the zero-mark others display a broader spread.





Conclusively, these figures offer a deep dive into the nuances of specific humidity retrieval differences when juxtaposed against a consistent background dataset. They underscore the variability introduced by different datasets or satellite sources. The consistent patterns across events also hint at systematic challenges or characteristics inherent to the regions or times of the events studied.




**Figure 15**. Comparison of specific humidity profiles retrieved by WEGC against ECMWF background data for each event. (**a**) Difference in specific humidity (q$_{wegc}$-q$_{ECMWF-b}$) plotted against altitude. (**b**) Mean specific humidity represented by a solid line, with the standard deviation (std) denoted by a dashed line.






**Figure 16.** Detailed assessment of specific humidity relative differences across different events. Each of the six plots represents an individual event, containing three distinct visualizations for every satellite present: (i) Relative Specific Humidity Difference (RSHD) in percentage, showcasing the discrepancies between retrieved and background values, (ii) Standard Deviation (std) of RSHD, highlighted to capture the variability in differences, and (iii) Interquartile Range (IQR) of RSHD emphasizing the middle 50 % spread of the differences.





## 5. Conclusion

The critical role of Atmospheric Rivers (ARs) in global moisture transport and precipitation dynamics necessitates accurate
water vapor measurements for both understanding and forecasting these phenomena. While the integrated water vapor content
(IWV) of ARs can be well measured with microwave and infrared sounders, the vertical structure is less well known. In this
study, we analysed if specific humidity profiles and IWV values from Global Navigation Satellite System Radio Occultation
(GNSS-RO) measurements provide additional information for the study of ARs, in particular regarding their vertical structure.
The retrieval of water vapor from GNSS-RO data requires background information, which is usually incorporated by the one-
dimensional variational method (1D-Var) that combine observations and background in an optimal manner. We compared data
from the COSMIC Data Analysis and Archive Centre (CDAAC), operated by the University Corporation for Atmospheric
Research (UCAR) in Boulder, Colorado with data from the Wegener Center for Climate and Global Change (WEGC) at the
University of Graz, Austria.

A significant part of our study addresses the performance of the 1D-Var systems at CDAAC and WEGC, particularly when
there are large differences between the RO observations and the background data. The different computations of tangent point
trajectories and reference points, especially when they are situated at the edges of AR events with pronounced humidity
gradient, can lead to the extraction of background profiles at very different locations with different humidity regimes.

Despite these differences, the 1D-Var method at both centers yields remarkably similar retrieved profiles. In particular in the
altitude range, where the 1D-Var scheme allows for a strong weighting of the observations. This consistency across different
centres highlights the 1D-Var system's robustness and adaptability in handling diverse atmospheric conditions, demonstrating
its capacity to produce reliable profiles despite varied and contrasting background data. However, it is clear that background
water vapor significantly shapes the retrieved profiles, especially where weighting of the observations is weak.

When comparing data from different centres it is important be aware of the different computations of tangent point trajectories
and reference points, in order to better understand potential differences.

GNSS-RO data provide accurate specific humidity profiles, however, challenges in reaching the lowermost troposphere, lead
to systematic underrepresentation of the IWV values. The altitude-specific assessments shed light on the variability of IWV
data and the performance of RO profiles across different atmospheric layers. Additionally, the integration of Special Sensor
Microwave Imager/Sounder (SSMI/S) IWV values offered a broader perspective, revealing gaps in RO profiles and suggesting
the potential for augmenting RO IWV measurements with SSMI/S observations.



Comparing forecast data from the European Centre for Medium-Range Weather Forecasts profiles (ECMWF) with ECMWF analysis data at the GNSS-RO profile locations shows generally very small changes – even though the GNSS-RO profiles have been assimilated into the analyses. This indicates that GNSS-RO humidity information has still a limited impact on analyses – even though they have a demonstrated information content. It seems therefore useful to employ GNSS-RO (and other) data for direct observations of ARs.


As a next step, we thus recommend combining RO profiles with SSMI/S IWV observations to compensate for the missing lower parts of the RO profiles. While SSMI/S provides integrated water vapor data with horizontal resolution, it lacks vertical detail, making it a complementary dataset to RO profiles, especially near the edges of ARs where humidity gradients are most pronounced.


In conclusion, our research offers a comprehensive exploration of humidity profiles and IWV values within AR events. It illuminates the complex interplay between retrieval methodologies, tangent point trajectories, and background data, and their collective impact on the retrievals. Our findings not only advance the understanding of atmospheric moisture profiling but also set a direction for future research. Emphasizing the synergy of different observational tools and datasets, this study advocates
for an integrated approach to achieve a more accurate and comprehensive understanding of atmospheric conditions, particularly focusing on the dynamic nature of Atmospheric Rivers.

**Code availability:** The code used to produce the results of this study is available from the corresponding author upon qualified request.


**Author Contributions:** This article was jointly conceptualized by both authors. UF provided guidance, engaged in critical discussions, and oversaw the research process. The first author, BR, performed the computational and analytical work and was responsible for writing the manuscript draft, which was then revised by UF. UF's mentorship and expertise contributed to shaping the research direction and methodology. Both authors contributed to the interpretation of the results.


**Competing interests:** The authors declare that they have no conflict of interest.

**Acknowledgment:** the authors thank Mark Schwärz and Andrea K. Steiner (Wegener Center for Climate and Global Change, University of Graz, Austria) for invaluable discussions.


**Financial Support:** this work was funded by the Austrian Science Fund (FWF) under Research Grant W1256 (Doctoral Programme on "Climate Change: Uncertainties, Thresholds and Coping Strategies"), and by the University of Graz.





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
