# Peer review of "The Potential of Observing Atmospheric Rivers with GNSS Radio Occultation"

_Atmospheric Measurement Techniques, 2024_

## Referee Comment (RC2)

Review on "Observing atmospheric rivers using GNSS radio occultation data", by Bahareh Rahimi and Ulrich Foelsche

General comments.

In this paper, the authors analyze specific humidity profiles and IWV values from RO data measurements for the study of atmosperic rivers (ARs), focusing on their vertical structure. They consider data obtained at CDAAC and at WEGC, concluding that GNSS-RO data provide indeed additional vertically-resolved information, which was not already contained in the background or in operational analyses. IWV values from CDAAC and WEGC tend to underestimate the SSMI/S data, since GNSS-RO profiles not always reach the lowermost part of the atmosphere. The authors suggest that is promising to combine the GNSS-RO data (high vertical resolution) with SSMI/S data (high horizontal resolution) to get a more compete view of the 3D structure of ARs.

This is an interesting intercomparison between different techniques with data processed at different centres. Some recommendations are made, mainly for obtaining specific humidity profiles in the lower troposphere and in relation to the different calculations of tangent point trajectories and reference points. The authors conclude that their results contribute to the understanding of atmospheric moisture profiles and set a direction for future research. It is not clear to me what that specific direction is for a more accurate and complete understanding of ARs.

In an AR the moisture is transported along narrow corridors, often driven by large-scale weather patterns like cyclones. The process typically starts when warm ocean waters evaporate and the resulting moisture is lifted into the atmosphere. When these rivers encounter land or mountains, the moisture condenses, leading to heavy rainfall or snowfall. Therefore, it would be very illustrative to indicate the synoptic conditions corresponding to each case study.

On the other hand, we know that mesoscale models, like WRF, have been instrumental in advancing our understanding of ARs, particularly with regard to their dynamics. There are numerous studies on ARs using mesoscale models. These are capable of simulating meteorological phenomena on scales ranging from a few km to hundreds of km and are particularly valuable for understanding the detailed structure and dynamics of ARs. In particular, they are able to resolve important features of ARs, such as their interaction with topography, the development of precipitation bands and the processes leading to extreme precipitation and flooding. I would then suggest that the authors indicate to what extent the results presented in these case studies help to understand, through the combination of GNSS-RO data with SSMI/S, data, the structure and dynamics of ARs and the processes leading to extreme rainfall and flooding.

The results obtained in this paper seem to be useful in a more general context and not particularly pertinent to RAs. In summary, what can be concluded about the generation and evolution of RAs from RO data in addition to satellite data that cannot be inferred or forecasted from mesoscale models?

Scientific significance: Good. Scientific quality: Good. Presentation quality: Good.

The paper address relevant scientific questions within the scope of AMT. It presents novel concepts. Additional conclusions could have been reached, specifically to the knowledge of the dynamics of the atmospheric rivers. The scientific methods and assumptions are clearly outlined. The description of experiments and calculations are sufficiently complete and precise to allow their reproduction. The authors give proper credit to related work and clearly indicate their own contribution. Perhaps the seminar paper by Bevis et al (JGR, 1992) could have been included too. The title should include both complement techniques, not only RO data. The abstract provide a concise

summary. The overall presentation is well structured. I am not native so I cannot comment about the English language. The mathematical formulae, symbols, abbreviations, and units are correctly defined and used. The references are appropriate.

Line 314 > Figure 3(d).

---

## Author Response (AR1)

We would like to thank the first reviewer for the positive assessment of our work and for the constructive comments and questions. We will revise the manuscript accordingly. Please find our responses to all comments below:

**General comments:**

**1. Introduction: is there any existing literatures showing high vertical resolution moisture retrieval, instead of just IWV, can significantly improve the AR forecast? If so, it will be more convincing to include them.**

Ans: Thank you for this suggestion. There is indeed literature indicating that high vertical resolution moisture retrieval, particularly from GNSS-RO data, can improve the forecasting of Atmospheric Rivers (AR). Studies such as Ma et al. (2011) and Neiman et al. (2008a) have shown that the assimilation of vertical moisture profiles from GNSS-RO improves AR intensity and trajectory predictions.

- *Ma et al. (2011)* demonstrated that the inclusion of high-resolution GNSS-RO data into weather models significantly improved AR landfall predictions.
- *Neiman et al. (2008a)* showed the benefits of GNSS-RO profiles in observing AR structures, particularly highlighting their vertical moisture transport.

We have already cited these papers in the introduction (lines 84, and 85); however, we will include further references in the revised manuscript (The modification is on Line 92-101.)

2. I agree that combining the MWR and RO can better resolve the water vapor in both dimensions and compensate the missing data gap of RO at lower troposphere, but it is not directly related to the work presented in the manuscript. It will be good to strengthen the link between these two and state why this is a good idea based on the results you get. For example, "the consistent RO and SSMI/S IWV values shown in this manuscript demonstrate the possibility of combining these two observations within the current variational method framework".

Ans: Thank you for your suggestion and the precise formulation, we will be happy to include it in the revised manuscript (line 851-852).

**3. It's better to point out which physical observation is used in 1DVar for each center: refractivity, bending angle, or phase delay?**

Ans: Thank you for pointing this out. We will clarify in the methodology section which physical observations are used by each center in their 1DVar retrieval. Specifically, the University Corporation for Atmospheric Research (UCAR) employs bending angle data, while the Wegener Center (WEGC) uses refractivity profiles as input for their retrieval process. This information will be added to improve clarity (The modification is on Line 163-165).

4. It seems the sensitivity to the vapor a-priori is somewhat mixed – Fig. 4(a) is small, Fig. 6(a) is large. Some lines stress the insensitivity of the retrieval from various backgrounds (L21, L352, L726), others saying the opposite (L623, L656). I suggest having a consistent, clear view on this to avoid confusing the readers. Besides, RO data could be biased (especially in the lower troposphere as shown in Fig. 15 and 16 at 0-2 km) too. This factor should also be included in the discussion of background/data impacts on 1DVar.

Ans: Thank you for your observation. Please see also our answer to your minor comment 1 below. Which should help to clarify this.

The statements in (L21, L352, L726) refer to humidity profiles, and to the altitude range, where the RAER is less than 70 %, and where the retrieved profiles therefore can considerably diverge form the background. Here, we find good agreement between WEGC and CDAAC, even when the background profiles are very different. The statements in (L623, L656) refer to integrated water vapor, and L623 is quite a special case, where CDAAC data for TSX agree quite well with the respective background (ECH) while CDAAC data for the other satellites are generally drier than their background profiles (ERA). Note that the retrieved IWV don't necessarily have to agree with the background IWVs. The modification is on Lines 765-767 and 770-775.

**Minor comments:**

1. Line 284: By this definition how do we interpret RAER when it is 100% (above 10km)? The RO observation uncertainty is extremely high in this altitude range and the retrieval mainly follows the background? It will be helpful to explain further on what RAER represents.

Ans: Thank you for your comment. Yes, you are correct. To improve the explanation, we will change the paragraph (after line 291 in old version) to:

A RAER value of 100 % means that the RO observation uncertainty is extremely high and that the retrieved profile at this altitude is identical to the background profile. At altitudes with RAER > 70 % the retrieved profile is still be background-dominated, this is usually the case above approximately 9 km at low latitudes and above 4 km at high latitudes during winter - but also in the lowest few hundred meters, where the RO refractivity profile can be biased due to superrefraction. At altitudes with RAER < 70 % the retrieved profile is observation-dominated and we can expect significant differences to the background profile. This additional explanation has been added on line 325-330.

**2. Line 314: Should be Fig. 3(d) instead of Fig. 3(e) ?!**

Ans: Sorry for this mistake and thank you for catching this error. We will correct the reference to Figure 3(d) instead of Figure 3(e). We have addressed this issue in Line 347 of the updated document

**3. Line 459: Interesting results. Both centers should use the same data - is there a BA or N retrieval difference between the centers?**

Ans: There are differences in the BA and N retrievals, but the different 1DVar techniques with different background data play a larger role. However, the identified IWV differences are not very large. Additional explanation has been added on line 506-510 of the updated manuscript.

**4. Line 472: period between "stable" and "Here".**

Ans: Thank you, we will correct the typographical error. The change has been made in Line 523 of the revised manuscript.

**5. Line 546: What is the anticipated IWV considering the missing lowest 200m of RO data? What could be the possible cause? This is maybe an open question but I suggest to discuss it if it is being mentioned.**

Ans: Since RO profiles generally don't reach all the way down to the surface, we have to expect a systematic underrepresentation of the "true" IWV content - as discussed in the manuscript. The "200 m" don't have a particular meaning, we had to find a balance between ensemble size and penetration depth. There are also profile reaching further down, but not enough for meaningful statistics. As per your suggestion, we have updated the text in Line 598-603.

**6. Line 578: What is the reason of different numbers? Is it because the cut-off height very different between UCAR and WEGC?**

Ans: Yes, you are correct. The cut-off heights for RO profiles differ between UCAR and WEGC due to variations in retrieval methods and different quality controls applied. This distinction results in differences in the number of events recorded by each center. In this specific case, the number of events in CDAAC and WEGC is distinct, largely due to these cut-off height differences. Additional explanation has been added on line 638-641 of the updated manuscript.

7. Line 624: "This emphasizes the importance of the choice of background profiles and how they can influence the retrieved IWV values": Not sure if this is a correct statement. Based on the results it seems ERA has a wet bias, and CDAAC retrieval bring it back to truth so they are drier than ERA. When ECH is used without the wet bias, CDAAC retrieval matches them better. So no matter which background profile is used the retrieved IWV is statistically not sensitive to it, is it correct? If so the background profiles should influence the IWV bias compared to the background, rather than the retrieved IWV. I recommend clarifying the sentence.

Ans: Thank you for highlighting this point. We will revise this statement to accurately reflect that the retrieved IWV values themselves are not highly sensitive to the choice of background profile, but rather the background profile influences the bias relative to the true IWV. This clarification will be made to ensure consistency with the results. We clarified it as requested. This change can be found on Line 685-688 in the revised manuscript.

**8. Line 625: Figure 13 instead 11?!**

Ans: Thank you, this was a mistake. We will correct the reference to Figure 13 instead of Figure 11. We have correct this in Line 689 of the updated document.

**9. Line 628: There is actually no CDAAC and WEGC IWV data comparison in the section 4.3.2.. And in section 4.2 this specific case (Iceland -UK 2009) is not shown. I suggested to add this case in 4.2 to validate the statement.**

Ans: We appreciate your attention to detail (and apologize for our mistake). There is a typographical error in the manuscript—the correct section is 4.3.1, where the Iceland-UK 2009 event is mentioned. In Section 4.2, we focused on discussing three out of the six AR events in detail, and given that the results for the Iceland-UK event were similar to the other events, we decided not to repeat them in Section 4.2. This is corrected in Line 692 of the updated document.

10. Line 655: It will be interesting to bring up the COSMIC-2 dry bias issue (Line 457) here again since it is the same case. In Fig. 14 C2 does not show an obvious bias w.r.t the background for both CDAAC and WEGC. However in Fig.7 C2 from CDAAC is obviously dryer than the one from WEGC. Does it indicate that the ECMWF-b is dryer than corresponding ECH profiles for C2?

Ans: We agree that it would be insightful to revisit the COSMIC-2 dry bias issue here. We will add a discussion comparing the bias observed in Figures 7 and 14, and attempt to explain why COSMIC-2 shows different bias characteristics between CDAAC and WEGC. The change has been made in Line 728-733 of the revised manuscript.

**11. Line 670: It will be clearer to define the RSHD with an equation.**

Ans: Thank you for this suggestion, we will include an equation defining RSHD in the revised manuscript. We clarified the methodology as requested. This change can be found on Line 746-750 in the revised manuscript.

**12. Line 731: should this sentence be in the same paragraph above?**

Ans: We will shorten this sentence and move it to the appropriate paragraph (starting at line 720 in the old version). The change is on line 819 in the updated manuscript.

\_\_\_\_\_

We thank the second reviewer for the constructive and detailed comments. We will revise the manuscript accordingly. Please find our responses below.

**1. Future Research Direction:** We appreciate your comment regarding the future research direction. In the conclusion, we emphasize that combining GNSS-RO data with passive microwave sensors like SSMI/S offers promising potential for advancing our understanding of ARs. However, we recognize the need for a more explicit discussion about future research directions. To clarify, future work should focus on integrating these complementary datasets within advanced data assimilation systems to better resolve the three-dimensional moisture structure of ARs, particularly in the lower troposphere where GNSS-RO data have limitations. Additionally, the impact of GNSS-RO data on improving AR forecasts and understanding extreme precipitation events could be further explored by combining the strengths of RO's vertical resolution and SSMI/S's horizontal coverage in real-time operational forecasting systems.

Another important aspect (also related to comment 3) is that GNSS-RO data are available globally, allowing to study ARs in areas, which are generally not covered by intense measurements campaigns or routinely studied with mesoscale models. As an example: In a current study we analyze ARs affecting Africa – a hitherto understudied region in terms of ARs, where GNSS RO data prove to be valuable. We will include this aspect in section 5, which we will furthermore rename to "Conclusions and Outlook". We have addressed this issue in Line 854-864, and line 804 of the updated document.

2. Synoptic Conditions for Each Case Study: Thank you for the suggestion to include the synoptic conditions corresponding to each case study. We agree that understanding the large-scale weather patterns driving AR events is important. While we acknowledge the synoptic context was not detailed in this paper (which is already quite long), each case study was selected based on specific AR events already known for their synoptic conditions (e.g., cyclones driving moisture transport). We will consider adding a more detailed synoptic overview in future work, but the primary focus of this manuscript is on evaluating the performance of GNSS-RO in measuring vertical and horizontal moisture distributions in ARs. This can set the foundation for further studies linking moisture distribution with synoptic patterns.

**3. Role of Mesoscale Models and GNSS-RO Data:** We appreciate the comment regarding mesoscale models like WRF and their ability to simulate ARs, including the interaction with topography and precipitation processes. Our study's goal is not to replace mesoscale models but to demonstrate the complementary value of GNSS-RO and SSMI/S data in capturing the vertical and horizontal structure of moisture in ARs. The additional vertically-resolved information from GNSS-RO, particularly in the mid and upper troposphere, can enhance mesoscale models by providing more accurate initial conditions in regions where in situ observations are sparse. The combination of GNSS-RO data with mesoscale models, particularly through

assimilation, could further improve our understanding of AR dynamics and the processes leading to extreme precipitation and flooding. This is an important direction for future studies. As per your suggestion, we have updated the text in Line 112-122.

**4. Usefulness of Results for AR Understanding:**

**Ans 1**: Our results demonstrate the usefulness of GNSS-RO data in contributing to a more complete picture of the moisture structure in ARs, particularly when combined with SSMI/S data. While the study does not directly focus on AR dynamics, the vertically-resolved humidity data from GNSS-RO can complement mesoscale model outputs, offering a way to improve moisture analysis in regions where models struggle with resolution or observational gaps. Specifically, the high vertical resolution of GNSS-RO data is valuable for capturing the moisture distribution in ARs, which is not fully resolved by passive radiometers or models alone. In terms of AR generation and evolution, GNSS-RO data provide valuable independent observations that could help verify or refine the moisture profiles simulated by mesoscale models, improving forecast accuracy.

**Ans 2:** Thank you for raising this important question. GNSS-RO data, particularly due to its high vertical resolution, offers unique insights into the moisture structure of atmospheric rivers (ARs) that cannot always be captured by mesoscale models or passive satellite sensors. One specific contribution of our study is the ability to detect sharp gradients in Integrated Water Vapor (IWV) at the edges of ARs, which we explain in the paper. These sharp gradients are often critical for understanding the transition between moist and dry regions within ARs.

Our results show that the two data centers (CDAAC and WEGC) provide differing insights into these gradients, which highlights the complexity of moisture retrieval in these regions. This additional layer of information on moisture variability, particularly in the vertical structure, can complement mesoscale models like WRF, which may struggle to fully resolve these sharp transitions, especially in the lower troposphere. Thus, GNSS-RO data adds valuable detail about AR evolution that might not be fully captured by traditional models and satellite observations alone. As per your suggestion, we have updated the text in Line 112-122.

**5. Reference to Bevis et al. (1992):** Thank you for bringing up the work by Bevis et al. (1992). We agree that this paper is highly relevant, particularly regarding the retrieval of water vapor using GPS techniques, and could be cited to further strengthen the introduction. This foundational work sets the stage for understanding how GNSS-RO contributes to moisture retrieval, and we will consider citing it in future work. The change has been made in Line 57-61 of the revised manuscript.

**6. Title Suggestion:** We appreciate your suggestion regarding the title. This paper will be part of special issue on "Observing atmosphere and climate with occultation techniques ...", therefore we initially thought that the paper title would be appropriate. In light of this comment (and also the comment 4 above) we will change the title to:

"The Potential of Observing Atmospheric Rivers with GNSS Radio Occultation".

**Specific Comments**

1. Line 314 (Figure Reference): Thank you for catching this mistake. We will correct the reference to Figure 3(d) instead of Figure 3(e). We have addressed this issue in Line 347 of the updated document